# Evaluation of Bone Repair Using a New Biphasic Synthetic Bioceramic (Plenum^®^ Oss_hp_) in Critical Calvaria Defect in Rats

**DOI:** 10.3390/biology12111417

**Published:** 2023-11-10

**Authors:** Paula Buzo Frigério, Lilian Caldas Quirino, Marisa Aparecida Cabrini Gabrielli, Pedro Henrique de Azambuja Carvalho, Idelmo Rangel Garcia Júnior, Valfrido Antonio Pereira-Filho

**Affiliations:** 1Department of Diagnosis and Surgery, Araçatuba School of Dentistry, São Paulo State University (UNESP), Araçatuba, São Paulo 16015-050, Brazil; irgcirurgia@gmail.com; 2Department of Diagnosis and Surgery, Araraquara School of Dentistry, São Paulo State University (UNESP), Araraquara, São Paulo 14801-903, Brazil; liliancqodonto@yahoo.com.br (L.C.Q.); marisa.gabrielli@unesp.br (M.A.C.G.); carvalhopha@gmail.com (P.H.d.A.C.); valfrido.pereira-filho@unesp.br (V.A.P.-F.)

**Keywords:** bioceramics, bone substitutes, bone grafting, biomaterials, osteoinduction, animal models

## Abstract

**Simple Summary:**

The search for regenerative treatments is one of the main objectives of researchers to find effective ways of repairing bone defects and forming areas of bone of sufficient quality and quantity which can receive osseointegrated implants to reconstruct areas affected by periodontal lesions and/or systemic diseases. Biphasic bioceramics are synthetic bone substitutes made of hydroxyapatite (HA) and β-tricalcium phosphate (β-TCP). These materials have osteoconductive properties and are chemically and structurally similar to bone matrix, providing better safety and predictability during guided bone regeneration. This study aimed to evaluate the bone repair process using a new biphasic bioceramic (Plenum^®^ Oss_hp_—70HA: 30β-TCP) compared to a synthetic graft (Straumann^®^ BoneCeramic™) in a critical defect in the calvaria of rats. It was observed that the new Plenum^®^ Oss_hp_ bioceramic, in the rat calvaria model, offered excellent safety and surgical predictability, being efficient in the bone repair process. Plenum^®^ Oss_hp_ behaved similarly to the commercial control (BoneCeramic™) and can be considered an effective option as a synthetic bone substitute in the dental clinic, especially for implantology.

**Abstract:**

(1) Background: Biphasic bioceramics are synthetic bone substitutes that provide greater safety and better predictability in guided bone regeneration. This study aimed to evaluate the bone repair process using a new biphasic bioceramic of synthetic origin (Plenum^®^ Oss_hp_—70HA: 30β-TCP) in critical calvarial defects. (2) Methods: seventy-four defects were created in rat calvaria and divided into two groups—Plenum^®^ Oss_hp_ (PO), right side, and Straumann^®^ BoneCeramic™ (BC), left side. Euthanasia was performed at 7, 15, 30, and 60 days after surgery. (3) Results: Lower gene expression was observed for runt-related transcription factor 2 (RUNX2) and vascular endothelial growth factor (VEGF) and higher expression for Integrin Binding Sialoprotein (IBSP). The results correlated with moderate immunolabeling for osteocalcin (OCN) and slight immunolabeling for osteopontin (OPN) in the PO group. Histometry showed a greater amount of biomaterial remaining in the PO group at 60 days. The microtomographic analysis showed a lower density of bone connectivity and a greater thickness of the trabeculae for the remnants of the PO group. (4) Conclusions: the Plenum^®^ Oss_hp_ showed no differences compared to BoneCeramic™ and is therefore considered an effective option as a synthetic bone substitute in bone regeneration.

## 1. Introduction

Critical bone defects are classified as those in which bone repair is not completed spontaneously, requiring the use of grafts for filling or reconstruction and simultaneous stimulation of bone neoformation [1]. 

Areas of extensive bone defects are still considered a challenge for dental practice [2]. The search for regenerative treatments is one of the main objectives of researchers, aiming to create techniques capable of repairing bone defects and forming areas of sufficient bone quality and quantity to receive osseointegrated titanium implants capable of supporting functional load for reconstruction of regions affected by periodontal and/or systemic diseases [1,2,3].

Autogenous bone grafts are the gold standard for guided bone repair and regeneration (GBR) due to their osteoinductive and osteogenic properties [3,4,5,6,7]. However, the disadvantages related to this technique, such as a limited amount of bone, a need for a donor site, infections, and post-surgical trauma, make the search for new alternatives increasingly requested [3,4,5,6,8].

In recent decades, efforts to search for efficient procedures and biomaterials for reconstructing bone defects in clinical dentistry have been extensively documented [1,5,7]. Synthetic bone grafts must meet specific prerequisites to develop healthy bone tissue formation, offering biocompatibility and safety, thus being free from the risk of infection [2,5]. Living tissues must be able to modulate inflammatory responses and accept the properties released by synthetic biomaterials [2].

Among the classes of alloplastic regenerative materials, which are alternatives to autogenous bone, we have bioceramics, developed to serve as a mineralized framework for the proliferation of osteoprogenitor cells [1,5]. These biomaterials can influence osteoblasts’ viability and activity mainly because they have a chemical and physical composition resembling bone tissue [2]. Bioceramics are composed of hydroxyapatite (HA) and β-tricalcium phosphate (β-TCP), thus indicating their use as efficient bone substitutes [1,9,10,11,12,13,14,15].

Hydroxyapatite has similarities to bone tissue and is widely applied as a grafting material in bone regeneration surgeries due to its excellent biocompatibility and osteoconductive characteristics [8,16,17,18]. Beta tricalcium phosphate is one of the most attractive bone repair materials for regenerative areas, has excellent osteoconductive and osteotransductive properties, and has a crystalline structure of simple degradation and thermal stability [1,19,20].

Due to the slow resorption rate, HA presents deficiencies in the bone neoformation process. However, it is capable of maintaining its structure and framework for a longer period [2,8,19]. On the other hand, β-TCP presents high solubility and rapid resorption, favoring bone formation but hindering the maintenance of the framework [2,11,15,19,21].

In the study conducted by Fabris et al. in 2018 [22], which evaluated the osteoconductive potential of BoneCeramic™ in critical calvarial defects in rats, compared with autogenous bone and clot, it was observed that there was greater bone formation with the use of autogenous bone in the 28-day period, and in the 14-day period, there was no difference. Thus, the authors concluded that the biomaterial tested was favorable for filling bone defects but not superior to autogenous bone graft.

Lee et al., in 2013 [12], evaluated the use of a bioceramic composed of β-TCP:HA, in the proportion of 60%:40%, compared to Bio-Oss^®^ in critical defects in rat calvaria. They observed better bone neoformation in the group where the bioceramic was applied, with an increase in bone formation and volume in the central region of the defect when compared to the defects filled with Bio-Oss^®^, concluding that the bioceramic obtained a superior bone formation, both in quantity and quality, in the studied model, suggesting its use as an effective bone substitute.

There is still no consensus on the proportion of bioceramics most suitable for bone grafting procedures, and this concentration should be adjusted according to clinical conditions. For patients who require the maintenance of the framework for a longer time, the proportion of HA should be higher than that of β-TCP. In contrast, for cases where a framework exists and requires a higher rate of bone formation, the proportion of β-TCP should be higher than HA [15,19]. In support of those findings, it is known that the union of β-TCP to HA provides osteoinductive effects and the promotion of bone neoformation in animal models which were not evident when either material is used alone, and the junction of the two ceramics demonstrates their biofunctionality [3].

The new Plenum^®^ Osshp bioceramic (M3 Health Ind. Com. de Prod. Med. Odont. e Correlatos S.A, Jundiaí, Brazil) is composed of hydroxyapatite and β-tricalcium phosphate in a ratio of 70% HA/30% β-TCP. The combination of these two biomaterials overcomes their disadvantages, creating a potential alternative for guided bone regeneration in critical defects that do not allow for the possibility of other means of grafting [5].

Thus, the objective of this study was to evaluate the bone repair process using a new biphasic bioceramic (Plenum^®^ Oss_hp_-M3 Health Ind. Com. de Prod. Med. Odont. e Correlatos S.A, Jundiaí, Brazil) compared to a synthetic graft (BoneCeramic™-Straumann Holding AG, Peter Merian-Weg 12, 4002 Basel, Switzerland), regarding the analysis of gene expression (RT-PCR), computed microtomography, and histometry and immunohistochemistry in the filling of critical defects in rat calvaria. 

## 2. Materials and Methods

### 2.1. Experimental Design

This study was approved by the Ethics Committee on Animal Use (CEUA) of the Araraquara School of Dentistry, with registration number 43/2017. The experiments were performed according to the standards established by the ISO 10993 standard—“Biological evaluation for medical devices—Part 6: Tests for local effects after implantation” [23].

To create the critical defects in the calvaria, 37 adult male rats (Rattus Norvegicus Holtzman) weighing approximately 350 g, with an average of 3 months of life, provided by the Animal Facility of the São Paulo State University—UNESP/Araraquara, were used. The animals were kept under standardized conditions, with water and food ad libitum and under room temperature of 22 ± 2 °C, with 12-h light/dark cycles.

The sample consisted of 74 critical defects made in rat calvaria. In each rat calvaria, two critical defects were created and filled with different materials that constituted two groups: Group PO—Plenum^®^ Oss_hp_ (M3 Health Ind. Com. de Prod. Med. Odont. e Correlatos S.A, Jundiaí, SP, Brazil) and Group BC—Straumann^®^ BoneCeramic™ (Straumann, Basel, Switzerland), filling the right and left defect of the rat calvaria, respectively.

### 2.2. Randomization and Sample Size

The sample size calculation was based on data from a study on critical defects in the calvaria of rats. In this experiment, 24 rats were divided into three groups, which was sufficient to carry out comparative analyses of the process of bone formation, maturation, and mineralization [24]. Thus, for the present study, the power of the test was calculated, giving a power of 0.8 and a type 1 error of 0.05. Based on these experiments, we calculated that 37 animals comprising 74 calvarial defects, group PO and group BC, respectively, right and left rat calvarial defect, would be sufficient to reveal statistical differences in this study. We used *n* = 8 samples for immunohistochemistry and histometry in the 7-day period, *n* = 5 samples for RT-PCR in the 15-day period, *n* = 8 samples for histometry in each of the 15- and 30-day periods, and *n* = 8 samples for Micro-CT in the 60-day period, so *n* = 37 samples for each of the groups evaluated.

### 2.3. Surgical Procedure (Critical Calvarial Defect)

The animals were anesthetized by intramuscular administration of ketamine hydrochloride at a dose of 30 mg/kg (Syntec—Santana de Parnaíba, SP, Brazil) and xylazine hydrochloride at a dose of 10 mg/kg (Syntec—Santana de Parnaíba, SP, Brazil) [25]. After anesthesia and trichotomy, the animals received a V-shaped incision over the parietal bones, followed by detachment. The two bone defects in the calvaria of the animals were made with the aid of a 5 mm internal diameter trephine (Ref 103.027, Neodent System, Curitiba, PR, Brazil) under constant irrigation of 0.9% saline, and consisted of full-thickness perforation of the parietal bone, maintaining the integrity of the dura mater [24].

After filling the defects, right side (Plenum^®^ Oss_hp_) and left side (Straumann^®^ BoneCeramic™) (Figure 1) sutures were performed with 5-0 mononylon (Ethicon^®^, Johnson & Johnson, São José dos Campos, SP, Brazil).

All animals received a single intramuscular dose of Dipyrone Sodium D 500 (Zoetis^®^, Campinas, SP, Brazil) and pentabiotic for small animals, 12,000UI (Zoetis^®^, Campinas, SP, Brazil) in the immediate post-operative period, and all evolved without signs of infection or other post-operative complications.

### 2.4. Euthanasia and Sample Separation

The animals were euthanized by anesthetic overdose at 7, 15, 30, and 60 days post-operatively, and the samples were collected. These were distributed according to the analyses to be performed: RT-PCR, using the antibodies runt-related transcription factor 2 (RUNX2), Integrin Binding Sialoprotein (IBSP), and Vascular Endothelial Factor (VEGF); micro-computed tomography (Micro-CT); histometric analysis (HE stain) and immunohistochemistry, for the proteins RUNX2, osteocalcin (OCN), and osteopontin (OPN).

First, the calvaria collected at the time of euthanasia were fixed in 4% paraformaldehyde solution for 48 h and washed in running water for 24 h, except for the samples for RT-PCR analysis, which were immediately added to cryotubes and stored in a −80 °C freezer until the time of analysis.

### 2.5. Gene Expression Analysis (RT-PCR)

The animals of the 15-day period were anesthetized by intramuscular administration of ketamine hydrochloride at a dose of 30 mg/kg (Syntec—Santana de Parnaíba, Brazil, SP) and xylazine hydrochloride at a dose of 10 mg/kg (Syntec—Santana de Parnaíba, Brazil, SP) [25]. Then, bone tissue was collected from the critical defects of 5 animals destined for the analysis of the expression of runt-related transcription factor 2 (RUNX2), Integrin Binding Sialoprotein (IBSP), and Vascular Endothelial Factor (VEGF) genes in the PO and BC groups.

Each bone fragment was removed using a 7 mm diameter trephine (Harte^®^, Ribeirão Preto, SP, Brazil) (Figure 2). Removal was performed under constant irrigation of a cooled phosphate-buffered saline (PBS) solution and subsequently packed in 2.0 mL cryotubes (Corning^®^, Corning, NY, USA), frozen in liquid nitrogen, and stored in a −80 °C freezer (Thermo Scientific, Waltham, MA, USA) so that total RNA could be isolated using the Trizol reagent (Life Technologies: Invitrogen, Carlsbad, CA, USA). RNA extraction was performed using the Promega kit (Promega Corporation, Madison, WI, USA).

RNA was quantified on a NanoDrop^®^ 2000 spectrophotometer (Thermo Scientific NanoDrop Technologies, Wilmington, DE, USA), and cDNA was prepared using 1 µg RNA by reverse transcriptase reaction (M-MLV: high-capacity reverse transcriptase (Waltham, MA, USA)). The primer sequence was amplified, and its products were subjected to electrophoresis with 1.5% agarose gel stained with ethidium bromide and visualized using Quantity One software (Version 4.5, Bio-Rad Laboratories, Philadelphia, PA, USA).

Finally, real-time PCR was conducted on the StepOne system (Applied Biosystems, Foster City, CA, USA) using the SybrGreen fluorescent detection dye system (Primer Design Ltd., Southampton, Hampshire, UK) under the cycling conditions of 95° for 10 min for primary denaturation, 35 cycles of 95° for 15 s for strand opening, and finally 60° for 60 s for primer annealing and fragment extension.

Gene expression was calculated by reference to mitochondrial ribosomal protein expression and normalized by gene expression of the defect wall fragments over the 15-day period.

### 2.6. Computerized Microtomography (Micro-CT)

The pieces from the 60-day period were stored in 70% alcohol for scanning in the microtomograph. For image acquisition, a spatial resolution of 12 μm was adopted; the images were acquired with a Phoenix V|tome| × M Microfocus Scanner (General Electric Company^®^, Wunstorf, Hanover, Germany), with a voltage setting of 80 kV. After image acquisitions, the axial, coronal, and sagittal planes were aligned to identify all regions of interest visually. Volumes of interest (VOIs) were acquired by reconstructing two-dimensional (2D) images, and 3D volumetric image reconstruction was obtained using SkyScan software, DataViewer^®^ v. 1.4.1.

In the CTAnalyser^®^ (SkyScan v. 1.11.8.0), a 5.0 mm circumference was positioned in the center of the defect areas, and a thickness of 1.0 mm was considered for analysis. A threshold of 105 (maximum) and 75 (minimum) was adopted for the evaluation of bone tissue, and 255 (maximum) and 105 (minimum) for the evaluation of regenerative materials. The parameter values for the regenerative materials were subtracted from the values obtained for the bone tissue to separate the volumes of the remaining regenerative materials from the newly formed bone tissue (Figure 3).

The following parameters were used for the dimensional analysis of trabecular bone microarchitecture, based on the nomenclature standardized by the American Society of Bone Mineral Research (2010) [26]: percent bone volume (BV/TV), trabecular number (Tb.N), trabecular separation (Tb.Sp), trabecular thickness (Tb.Th), and connectivity density (Conn.Dn).

### 2.7. Histometry Analysis (HE)

After microtomography, the pieces were decalcified in 10% EDTA for 4 weeks and verified by the correct demineralization; 6 µm thick sections of each piece were obtained and divided into slides with 4 sections each, stained by hematoxylin–eosin (HE).

Histometric evaluation of the specimens was performed at 15, 30, and 60 days using a DIASTAR light microscope (Leica Reichert & Jung Products^®^, Wetzlar, Hessen, Germany) with 5× magnification objectives. Representative images were sent to a microcomputer using a digital camera (Leica Microsystems^®^ DFC-300-FX, Wetzlar, Hessen, Germany), coupled to the ordinary light microscope, and saved under the TIFF extension. The values were determined using ImageJ FIJI^®^ image analyzer software (Broken Symmetry^®^ Software 1.43.76, Bethel, CT, USA). A trained examiner performed the analysis, taking the bone stumps as references and delimiting the defect. The area was delimited using the “freehand” feature in the software itself, and the total area of the samples was set as 100%. Then, the amount of bone tissue, biomaterial remnant, and connective tissue was measured using the color threshold tool, and the empty spaces were subtracted from the calculation when present (Figure 4). The percentage of each component was obtained by averaging duplicates of each defect.

### 2.8. Immunohistochemistry Analysis

Part of the decalcified specimens in 10% EDTA were used for immunohistochemistry analysis within 7 days. After a paraffin embedding processing, 6 µm histological sections were obtained. After deparaffinization, endogenous peroxidase activity was inhibited with hydrogen peroxide for 45 min. The slides were then subjected to antigen retrieval with phosphate–citrate buffer (pH 6.0) and endogenous biotin blocking using 1% bovine albumin and skimmed milk.

The primary antibodies used were against the proteins runt-related transcription factor 2 (RUNX2), osteopontin (OPN), and osteocalcin (OCN) (Santa Cruz Biotechnology, Dallas, TX, USA). Negative control was performed by omitting primary antibodies. As a positive control, the healthy cortical bone of the parietal bone outside the defect was used to verify the antibodies’ effectiveness. The secondary antibody used was a biotinylated anti-mouse antibody (Jackson Immunoresearch Laboratories, West Grove, PA, USA). Biotin avidin was used to amplify the immunolabeling signal, and the chromogen used was diaminobenzidine (Dako, Glostrup, Capital Region, Denmark). Samples were counterstained with Mayer’s hematoxylin solution. Images were obtained using a camera attached to a light microscope (Leica Reichert Diastar Products & Jung, Wetzlar, Hessen, Germany).

Descriptive analysis was performed by a single trained examiner (R.O), assigning scores to verify protein expression in the region of bone defects: (0) no labeling (0% cells/matrix), (1) light (<25% cells/matrix), (2) moderate (50% cells/matrix), and (3) intense (75% cells/matrix) [27,28,29].

### 2.9. Statistical Analysis

Statistical analysis was performed using GraphPad Prism software (GraphPad 8.01 Software, Inc.; San Diego, CA, USA). Micro-CT and histometry analyses were subjected to Shapiro–Wilk normality and Levene homoscedasticity tests, with more than 90% of the data showing normal distribution (Shapiro–Wilk, *p* > 0.05).

For the micro-CT analyses, the comparison of parameters at the threshold of neoformed bone tissue and biomaterial remnant was performed using the one-way ANOVA test. For histometry evaluations, the two-way ANOVA test and Tukey’s post-test were used.

The data from the gene expression analysis were submitted to the assumption of normality and homogeneity of variances by means of the Shapiro–Wilk and Brown–Forsythe tests, respectively, and presented values of *p* > 0.05 for normality and *p* < 0.05 for homogeneity. Statistical analysis was performed by comparing the mean and standard deviation (SD) values between groups.

## 3. Results

### 3.1. Gene Expression Analysis (RT-PCR)

The PO group (2.34 ± 0.13) showed lower RUNX2 expression compared to the BC group (2.93 ± 0.18). Lower VEGF gene expression was observed in the PO group (3.45 ± 0.21) compared to the BC group (4.93 ± 0.64). Bone sialoprotein (IBSP) expression was higher in the PO group (5.66 ± 0.32) compared to the BC group (2.79 ± 0.82) (Table 1).

### 3.2. Micro Analysis—CT

Regarding evaluation of the threshold of newly formed bone tissue at 60 days, in the PO group, there was a formation of mineralized tissue similar to the BC group in proportion to bone volume, as well as the number, separation, and thickness of trabeculae, but with a lower density of connections between them, with a statistical difference (Conn.Dn: PO = 26.74 ± 24.56 vs. BC = 69.12 ± 60.67) (Figure 5, Table 2).

As for the threshold for the biomaterial remnants, no statistical differences were observed in relation to the parameters BV/TV, Tb.N, Tb.Sp, and Conn.Dn, with a difference only for the thickness of the trabeculae (Tb.Th: PO = 0.32 ± 0.05 vs. BC = 0.16 ± 0.03) (Figure 5, Table 3).

### 3.3. Histometric Analysis

Newly formed bone tissue, connective tissue, and remnants of regenerative biomaterials were quantitatively analyzed as volumetric percentages (Table 4). The PO group did not show statistical differences compared to the BC group in the formation of connective tissue and bone (*p* > 0.05) for all of the periods evaluated (15, 30, and 60 days). When evaluated in relation to the biomaterial remnant, the PO group presented a higher percentage (0.78 ± 0.32) at 60 days compared to the BC group (0.4 ± 0.05) (Figure 6).

### 3.4. Immunohistochemistry Analysis

RUNX2 labeling was observed in cells characterized as pre-osteoblasts and young osteoblasts, quite close to the biomaterials and with a moderate pattern (2) for all groups.

Osteopontin was immunolabeled in both cells and the extracellular matrix. Light labeling (1) was found for OP and moderate labeling (2) for BC.

For osteocalcin, it was possible to observe the presence of moderate labeling (2) for the PO group. For the BC group, moderate labeling (2) was observed but with less evidence than when compared to the PO group (Figure 7).

## 4. Discussion

Synthetic bone grafts should be able to return sufficient bone quality and quantity to support prostheses on implants [30,31,32,33]. Today, a large proportion of surgeries requiring post-exodontic grafting or for the repair of small fenestrations, and bone dehiscences are performed using synthetic alloplastic grafts [32].

These synthetic bone substitutes are excellent because they are similar to bone tissue. Hydroxyapatite maintains the space and integrity of the defect, offering long-term maintenance, while β-TCP stimulates the formation of new bone through the release of calcium and phosphorus ions [30,31,32,33].

In order to define the best proportion between HA and β-TCP for reconstruction procedures, taking into account that a balance between bone resorption and deposition is needed for successful bone remodeling to occur, those biomaterials are produced and tested to determine their clinical use with greater predictability and safety [15,34,35]. Thus, this work tested a new biphasic bioceramic, comprised of 70% HA and 30% β-TCP, to verify its influence on bone healing [12,36,37,38].

Plenum^®^ Oss_hp_ was compared to a positive control biomaterial (Straumann^®^ BoneCeramic™—60HA: 40β-TCP) that is widely used in the field of surgery and implantology nowadays [39,40,41]. Thus, it was observed that Plenum^®^ Oss_hp_ is compatible as a bone substrate, since in the analysis of gene expression, there was osteoblastic differentiation during the repair process; in the comparison of mineralized tissue, there was no statistically significant difference with the leading product on the market (Straumann^®^ BoneCeramic™), and the osteoblastic cells showed activity in the bone mineralization stage in the PO group.

A lower expression of genes related to osteoblast differentiation was observed in the PO group. RUNX2 expression is present in pre-osteoblasts and is therefore an essential feature of bone neoformation at the beginning of the inflammatory process [42,43]. The OP group showed lower gene expression for bone neoformation when compared to the control group. However, it was observed that once synthesized, the new bone tissue shows greater mineralization and greater expression of bone sialoprotein (IBSP). IBSP has an osteogenic action and acts in the differentiation of bone marrow precursor cells into osteoblasts during the bone repair process [44].

Endothelial growth factor (VEGF) is derived from osteoblasts and acts directly in regulating the interaction between osteogenesis and angiogenesis, which is essential for the development and homeostatic maintenance of bone [45]. For angiogenesis, lower expression was also observed in the PO group compared to the BC group, a result similar to that published in 2014 by Chen et al. [46], who, when assessing angiogenesis in biphasic bioceramics with different proportions of HA and β -TCP, concluded that the higher the β -TCP phase, the higher the expression for VEGF, inducing an increase in the neovascularization of biphasic bioceramics. This statement corroborates the results of the present study, as the PO group has a lower β -TCP phase (30%), thus showing a lower expression of VEGF, with less invasion of blood vessels and migration of mesenchymal progenitor cells to the repair site [35,46,47]. This leads us to believe that the high proportion of HA (70%) leads to a lower rate of resorption and replacement by new bone, which impairs bone neoformation, since biological behavior is influenced by topography, particle size, and physical–chemical composition [48].

When comparing the mineralized tissue formed in the PO and BC groups, in the computed microtomography analysis, no statistical differences were observed between the volume of mineralized tissue, number of trabeculae, thickness, and separation between trabeculae, differing only in the connectivity density parameter (Conn.Dn), thus indicating a lower resistance of the newly formed bone tissue for the PO group. One hypothesis for this result is related to the difference between the proportions of its components, with the material tested having a more significant amount of HA compared to BoneCeramic™ (70:30 PO vs. 60:40 in BC). Although HA has a chemical composition resembling that of bone, it has greater fragility and less remodeling potential due to its low resorption rate after insertion, thus reducing the number of multiple connections [12].

In the study conducted by Lee et al. in 2013 [12], the authors evaluated a biphasic ceramic composed of 60 β-TCP:40 HA vs. Bio-Oss^®^ in critical defects in rat calvaria and observed, by microtomographic analysis, an increase in the BV/TV proportion in the biphasic bioceramic group when compared to Bio-Oss^®^, as well as a higher Tb.N and Tb.Th in the evaluated periods of 4 and 8 weeks, without statistical difference. The authors concluded that the better bone neoformation obtained by biphasic bioceramics is probably related to the superior osteoconductivity and osteoinductivity found in the biphasic composition of bioceramics compared to Bio-Oss^®^, which is in line with the results of our study which also showed similarities for BV/TV, Tb.N, and Tb.Th.

The immunohistochemical analysis performed after the 7-day period is of paramount importance, as it demonstrates the initial phase of the repair stage of the defect, a period in which there is a greater amount of cells that will be responsible for the synthesis of proteins which will participate in the reparational responses. Therefore, it is possible to evaluate the cellular response to the stimuli promoted by biomaterials. The RUNX2 protein is an important transcription factor that regulates osteoblast activity and mesenchymal cell differentiation, playing a critical role in bone formation [42,43,49]. RUNX2 labeling was moderate in both groups, with a normal bone neoformation process; so, at the beginning of the inflammatory process, the mesenchymal cells and pre-osteoblasts were differentiating normally.

Bone mineralization proteins are secreted by osteoblasts, OPN acts in the initial phase of bone mineralization, and OCN is a late marker of mineralization [50,51,52]. It was observed that the OP group presented a more advanced stage of bone mineralization in relation to the BC group for the OCN protein; however, both had moderate marking, a result that also corroborates the analysis of gene expression, with greater expression for IBSP [53], but the OP group had mild marking for the OPN protein. It was therefore possible to observe that the bone neoformation process was slow due to the low rate of HA resorption.

A critical point to be analyzed concerns the biomaterial remnant. Synthetic bone substitutes must have a controlled degree of resorption, as occurs in biphasic bioceramics, by balancing the rapid resorption of β-TCP and a lower rate of HA, preventing the biomaterial from being completely resorbed before bone neoformation occurs or remaining longer than necessary, impairing replacement by newly formed bone [12].

In the present study, when comparing the remnants of the PO group to the BC group by Micro-CT analysis, we observed similar volumes and the number, separation, and connectivity density of these particles, with greater thickness of the remnants in the PO group. The histological findings showed a greater amount of biomaterial remaining in the PO group only at 60 days. Nevins et al., in 2013 [54], when evaluating different formulations of biphasic bioceramics in the reconstruction of bone defects in an animal model, observed that the groups which had higher proportions of HA also had a greater amount of bioceramic remnant. These results suggest that, due to the difference in pore size and the ratio of bioceramics, the PO group, due to a greater amount of HA and larger particles, needs a longer time to be resorbed, presenting a biomaterial particle remaining with greater thickness than that of the BC group [54].

Takauti et al., in 2014 [55], compared Bio-Oss^®^, Endobon^®^, and BoneCeramic™ in critical defects in rabbit calvaria and observed that the synthetic regenerative biomaterial showed higher rates of newly formed bone tissue when compared to bovine biomaterials such as Bio-Oss^®^ and Endobon^®^. The present study observed that the PO group obtained similar results to the positive control (BC group).

In the near future, synthetic bone substitutes may be the gold standard of choice for GBR, replacing autogenous and xenogenous bone. Due to their osteoconductive and osteoinductive capacity, these materials have an excellent resorption rate and new bone formation rate, are safe and infection-free, are easy to handle, and, in addition, can be used with growth factors and/or cell transplantation [33].

Thus, in view of these findings, it was observed that the new bioceramic Plenum^®^ Oss_hp_, in the rat calvaria model, offered excellent safety and surgical predictability, being efficient in the bone repair process, and can serve as a synthetic bone substitute in clinical practice. However, because its particle size is larger than that of Straumann^®^ BoneCeramic™, this difference can be observed and the waiting time between resorption and replacement with new bone can be longer, thus generating differences between the biomaterials and the responses evaluated. More studies are therefore needed to assess the clinical effectiveness of alloplastic bone substitutes and to help dentists choose the ideal material.

## 5. Conclusions

Plenum^®^ Osshp proved to be effective in the bone repair process of critical defects in the calvaria of rats and behaved similarly to the commercial control (Straumann^®^ BoneCeramic™), thus being considered an effective option in histometric, immunohistochemical, and Micro-CT analyses as a synthetic bone substitute in the dental clinic, especially for reconstruction in implant dentistry. However, new percentages between the use of β-TCP and HA should be evaluated for a higher replacement rate of the biomaterial by bone, thus favoring bone formation.

## Figures and Tables

**Figure 1 biology-12-01417-f001:**
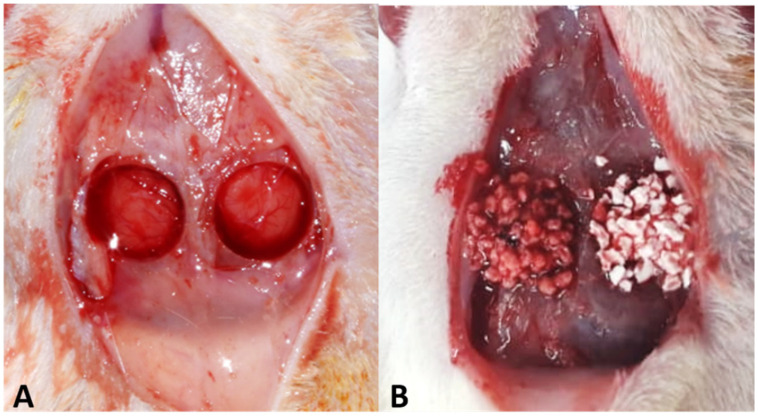
(**A**) Critical defects in calvaria, with 5 mm internal diameter. (**B**) The defect on the right side, group PO, was filled with Plenum^®^ Oss_hp_, and the defect on the left side, group BC, was filled with Straumamn^®^ BoneCeramic™.

**Figure 2 biology-12-01417-f002:**
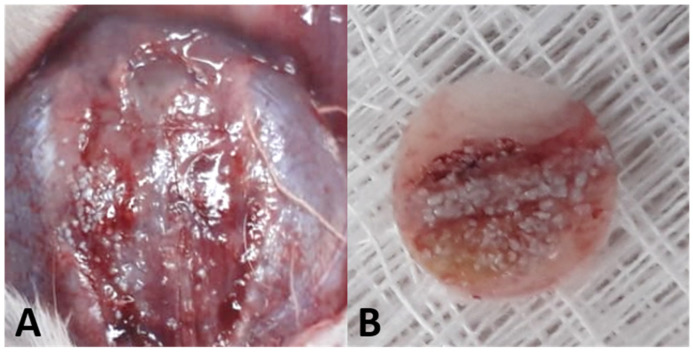
(**A**) Exposure of the calvarial region. (**B**) Bone fragment removed using a 7 mm trephine.

**Figure 3 biology-12-01417-f003:**
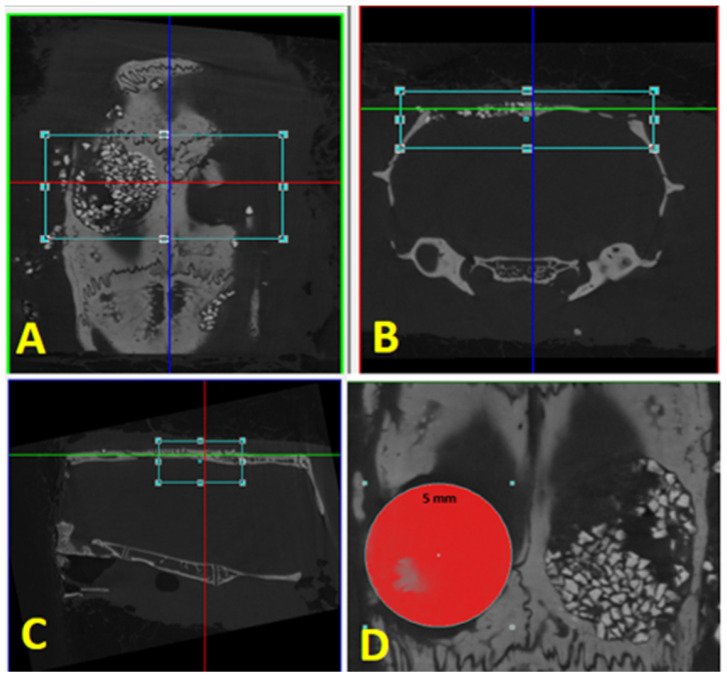
Reconstruction: (**A**) axial plane; (**B**) coronal plane; (**C**) sagittal plane. (**D**) Delimitation of the area of interest, with the same defect diameter of 5 mm.

**Figure 4 biology-12-01417-f004:**
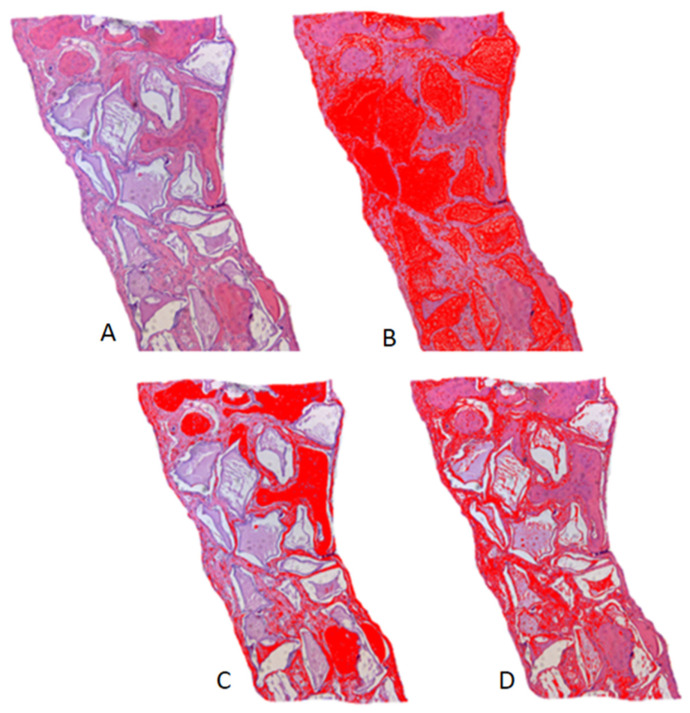
Histometry of the Plenum^®^ Oss_hp_ Group, 60-day period, illustrating the measurements by color threshold: (**A**) original histological image in H.E. staining; (**B**) threshold for biomaterial remnant; (**C**) threshold for bone tissue; (**D**) threshold for connective tissue. ImageJ FIJI^®^ software.

**Figure 5 biology-12-01417-f005:**
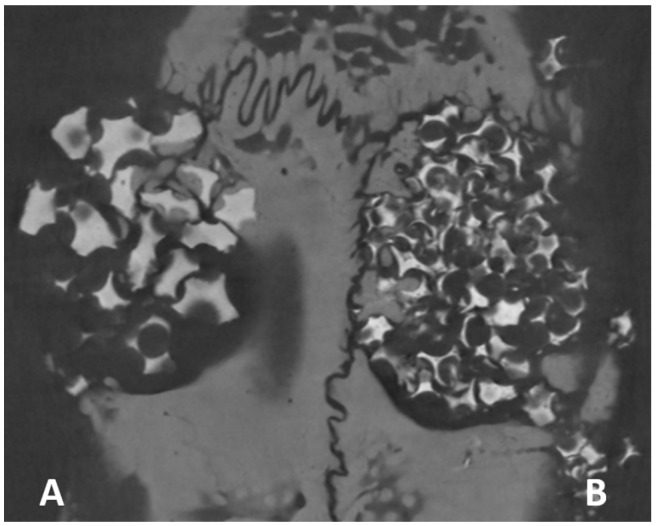
Images obtained by micro-computed tomography in the axial plane show the bone tissue permeating the biomaterial remnant. (**A**) Plenum^®^ Oss_hp_ and (**B**) Straumann^®^ BoneCeramic™.

**Figure 6 biology-12-01417-f006:**
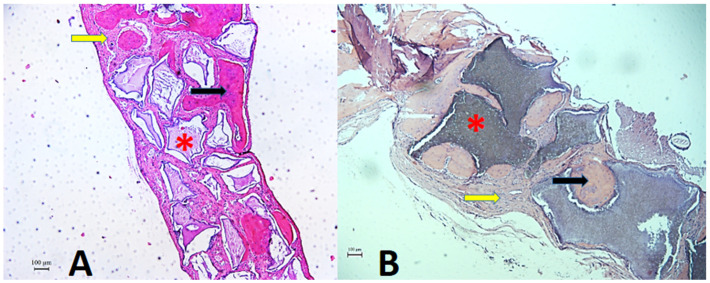
Histological section of the critical defect area in calvaria and adjacent regions. Photomicrograph showing histologic features of connective tissue (yellow arrow), bone tissue (black arrow), and remaining biomaterial (asterisk) at 60 days, post-operatively. (**A**). Plenum^®^ Oss_hp_ group. (**B**). Straumann^®^ BoneCeramic™. Staining: HE. Original magnification: 50×. Scale bars: 100 m.

**Figure 7 biology-12-01417-f007:**
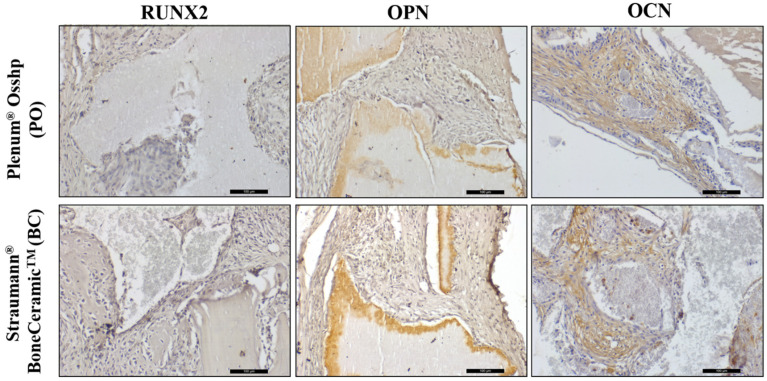
Immunohistochemical analysis showing the immunolabeling of RUNX2, OPN, and OCN proteins for the OP and BC groups at 7 days. Staining: Mayer’s hematoxylin. Original magnification: 20×. Scale bars: 100 m.

**Table 1 biology-12-01417-t001:** Gene expression analysis of RUNX2, VEGF, and IBSP for the analysis of regenerative biomaterials (PO vs. BC).

Expression Genes	Plenum^®^ Osshp (PO)	Straumann^®^ BoneCeramic™ (BC)
**RUNX2**	2.34 ± 0.13 ^a^	2.93 ± 0.18 ^b^
**VEGF**	3.45 ± 0.21 ^a^	4.93 ± 0.64 ^b^
**IBSP**	5.66 ± 0.32 ^a^	2.79 ± 0.82 ^b^

Note: Statistical differences are represented by lowercase letters (^a, b^). One-way ANOVA (*p* > 0.05).

**Table 2 biology-12-01417-t002:** Microtomographic parameters for mineralized tissue threshold.

Bone Tissue Parameters	Plenum^®^ Osshp (PO)	Straumann^®^ BoneCeramic™ (BC)
**BV/TV**	10.93 ± 3.00 ^a^	14.26 ± 4.19 ^b^
**Tb.N**	1.58 ± 0.35 ^a^	2.12 ± 0.61 ^b^
**Tb.Th**	0.07 ± 0.02 ^a^	0.07 ± 0.02 ^b^
**Tb.Sp**	0.47 ± 0.12 ^a^	0.38 ± 0.15 ^b^
**Conn.dn**	26.74 ± 24.56 ^a^	69.12 ± 60.67 ^b^

Note: Statistical differences are represented by lowercase letters (^a, b^). The difference for Conn.dn: PO = 26.74 ± 24.56 vs. BC = 69.12 ± 60.67. One-way ANOVA (*p* > 0.05).

**Table 3 biology-12-01417-t003:** Microtomographic parameters for the remaining biomaterial threshold.

Biomaterial Parameters	Plenum^®^ Osshp (PO)	Straumann^®^ BoneCeramic™ (BC)
**BV/TV**	25.63 ± 9.13 ^a^	15.8 ± 6.7 ^b^
**Tb.N**	0.80± 0.21 ^a^	0.96 ± 0.36 ^b^
**Tb.Th**	0.32 ± 0.05 ^a^	0.16 ± 0.03 ^b^
**Tb.Sp**	0.55 ± 0.13 ^a^	0.48 ± 0.15 ^b^
**Conn.dn**	4.64 ± 2.8 ^a^	8.80 ± 2.7 ^b^

Note: Statistical differences are represented by lowercase letters (^a, b^). The difference for Tb.Th: PO = 0.32 ± 0.05 vs. BC = 0.16 ± 0.03. One-way ANOVA (*p* > 0.05).

**Table 4 biology-12-01417-t004:** Histometric analysis of calvarial defects.

	Period (Days)	Plenum^®^ Osshp (PO)	Straumann^®^ BoneCeramic™ (BC)
**Connective Tissue**	**15**	0.82 (0.43) ^b^	0.87 (0.5) ^b^
	**30**	0.44 (0.17) ^a^	0.47 (0.26) ^a^
	**60**	0.80 (0.18) ^b^	0.75 (0.27) ^b^
**Neoformed Bone Tissue**	**15**	0.64 (0.3) ^a^	0.50 (0.19) ^a^
	**30**	0.45 (0.11) ^a^	0.28 (0.06) ^a^
	**60**	0.64 (0.3) ^a^	0.37 (0.21) ^a^
**Remaining Bone Gradt**	**15**	0.68 (0.3) ^ab^	0.64 (0.1) ^ab^
	**30**	0.56 (0.17) ^a^	0.40 (0.11) ^a^
	**60**	0.78 (0.32) ^a^	0.40 (0.05) ^b^

Note: Statistical differences are represented by lowercase letters (^a, b^). Two-way ANOVA and Tukey’s post-test (*p* > 0.05).

## Data Availability

The data presented in this study are available upon request from the corresponding author.

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
