# Peer review of "Evaluation of Bone Repair Using a New Biphasic Synthetic Bioceramic (Plenum^®^ Oss_hp_) in Critical Calvaria Defect in Rats"

_biology, 2023, doi:10.3390/biology12111417_

Round 1
Reviewer 1 Report
Comments and Suggestions for Authors
This manuscript is overall well written. However, there are some parts that need to be partially corrected.
1.“Introduction” is too long. I wish it was described more concisely.
2. # line 83-85 : delete
3. # line 106-108 : delete
4. At the beginning of “Discussion,” summarize and describe the key results of the paper.
5.Contents described in "Introduction" are repeated in "Discussion". Please reduce the overlapping content.
6.Insert your paper's shortcomings at the end of the "discussion".
Comments on the Quality of English LanguageOverall, there were no unnatural sentences while reading the manuscript. However, it would be a good idea to finally get it checked by an expert.
Reviewer 2 Report
Comments and Suggestions for Authors
The article titled “Evaluation of bone repair using a new biphasic synthetic bioceramic (Plenum® Osshp) in critical calvaria defect in rats” focuses on a comparative study between two commercially available biphasic bioceramics materials i.e., Plenum® Osshp and Straumann® BoneCeramic™. The manuscript is well written with rigorously experimental design, results, and discussion. However, the conclusion part seems very short, hence I would suggest the authors include all the important findings of each analysis in the conclusion part.
Reviewer 3 Report
Comments and Suggestions for Authors
Article with merit for publication and compatible proposal within the scope of the journal Biology. Point-to-point instructions for improving the text follow in the comments to the author. After moderate revisions required to the content of this version, the manuscript will be ready for publication.
2. Materials and Methods
2.1 Experimental Design
- Correct the term “ISSO” in line 121 with “ISO”. Rewrite the text of the citation, since the source is cited at the end of the article [23] and there is no need to place a link in this section of the manuscript.
- Choice of controls (lines 130 to 135): The study design did not mention a negative control, that is, a critical bone defect in the calvaria filled with the blood clot itself. The importance of its use would lie in affirming the osteoconductive efficiency of the biomaterial compared to an untreated site, even if the expected performance is not equivalent to BoneCeramic®. Was the ARRIVE animal reduction principle considered for this choice? Furthermore, BoneCeramic® has demonstrated promising results, but is not established in the literature as a gold standard in osteoconductive biomaterial such as Bio-Oss®, based on existing levels of evidence. Furthermore, BoneCeramic® is a two-phase ceramic in a ratio of 60:40 for HA and beta TCP, different from the ratio tested in Plenum® Osshp (70:30). Would the choice of positive control be guided by its alloplastic origin, credibility in the dental market? Therefore, the authors must better justify the reason for the study design and their options of not using the negative control and the option as a positive control to evaluate osteoconduction.
2.2 Randomization and Sample Size (lines and 136 to 144, 167 and 182):
A third group, BioGran®, was included in this section, whose results are not expressed or discussed in this article. If the intention was to justify the need for animal sampling, it is not clear why a bioglass was used for this standardization, different from the ceramic biomaterials tested. It appears to be a mistake and the authors must review the coherence of this information. Although the text considers the sample number of 24 animals, calculating 8 animals, 4 experimental times (7, 15, 30 and 60 days - line 167) and 2 bone graft therapies placed together in the same calvaria, the most correct would be to assume the number of 32 animals and not 37. In line 182 it was described that 5 animals were destined for gene expression analysis, so would there be only 3 animals left in each condition to evaluate the imaging and histopathological parameters? Or would these 5 animals be additional only for the time of gene expression at 15 days, and then add up to 32 and give a sample number of 37? This distribution is hardly understandable to the reader. Within the experimental design, it is foreseen: at 7 days immunohistochemistry and histometry which can be done with the same paraffin block, on the same animal; at 15 days genetic and histometric analysis which in general cannot be done on the same animal, as the first test is destructive and would require animals only for this purpose; at 30 days histometry and at 60 days microtomography and histometry which can be done with the same calvarial necropsy, one before demineralization/microCT and the other after/histology. If the actual quantity is 37 animals, the authors must clearly describe the constitution of the groups, experimental times, analytical techniques and number of animals for each condition, in order to justify the asymmetric number of experimental conditions and guarantee the reproducibility of the study.
2.3 Surgical Procedure (Critical Calvarial Defect)
The dose of 25mg/kg of ketamine hydrochloride is well below that recommended for the anesthetic association with xylazine. Authors should review the dose and route of administration in the animal management guidelines in force in the resolution n. 33/2016 CONCEA/MCTI/Brazil (https://www.gov.br/mcti/pt-br/acompanhe-o-mcti/concea/arquivos/pdf/legislacao/resolucao-normativa-no-33-de-18-de-novembro-de-2016.pdf/view) and describe them correctly in the manuscript.
2.6 Computerized Microtomography (Micro-CT)
Rewrite the text of the citation American Society of Bone Mineral Research (2010) on line 230, since the source is cited at the end of the article [25] and there is no need to link to this section of the manuscript.
3. Results and 4. Discussion
The authors must show the reader the relevance of the biomarkers RUNX2, VEGF, IBSP osteopontin and osteocalcin, together with the imaging and histometric results in the phenomenon of bone regeneration and compare the results found from the tested biomaterials with other evidence in the literature. Whether or not a low or moderate pattern is expected in each parameter depends on the experimental time. If there are biomaterials that promote a greater initial stimulus to osteodifferentiation (RUNX2), vasculogenesis (VEGF), organic composition that favors the nucleation of HA (OP and IBSP) or later collaborate in the more mature constitution of the bone organic matrix (OC). If the bone turnover is balanced with the rate of degradation of the granules and the limitation of fibrogenesis, with lower, equal or greater performances than other biphasic or monophasic alloplastic or bioceramic materials in accord to their histomorphometric evidence.
For a comparison with biphasic ceramics with different concentrations, I suggest adding the following reference, which worked with biphasic ceramics in the inverse proportion of HA and beta TCP to the article's proposal, 30:70, in a murine animal model and within the same condition in experimental study on cranial defects (Lomelino et al. 2012, https://doi.org/10.1007/s10856-011-4530-1). The morphological and histomorphometric data from this source will help to provide greater robustness to the discussion of the results found, debating the importance of HA and TCP in this context, as well as biocompatibility, as aspects of cellularity, tissue irritation and resorption were not explored in depth in this manuscript.
References:
The references are appropriate, in terms of quantity, quality and timeliness, in view of the theme proposed. Congratulations to the authors. It would only be necessary to deepen the topics described earlier in this opinion.
Tables and Figures:
Tables and figures are appropriate, in terms of quantity and quality. Only small corrections are made:
- in tables 1, 2, 3 and 4, format the statistical correlations, with superscript letters after the means and standard deviations and use the period instead of the comma, as the numerical notation must be expressed in the English language.
- in Figure 5, there is more material remaining in BoneCeramic (B) than in Plenum® Osshp (A). Would it be possible to substitute microCT images that are more representative of the tabulated findings of the aforementioned groups?
- Figure 6, appropriate scale bar, from 100m to 100 micrometers.
Round 2
Reviewer 3 Report
Comments and Suggestions for Authors
The authors provided adequate responses to all requirements identified point-by-point in the original text. Congratulations to the authors for the significant improvement of the content and format of the work. In this latest version, I consider the article suitable for publication.